# Adoption of mobile learning in the university context: Systematic literature review

**Alejandro Valencia-Arias**[1]*, **Sebastian Cardona-Acevedo**[2], **Sergio Gómez-Molina**[3], **Rosa María Vélez Holguín**[4], **Jackeline Valencia**[5]

1 School of Industrial Engineering, Universidad Señor de Sipán, Chiclayo, Perú, 2 Centro de investigaciones, Institución Universitaria Escolme, Medellín, Colombia, 3 Coordinación de Investigaciones e Innovación, Fundación Universitaria Católica del Norte, Medellin, Colombia, 4 Facultad de Ciencias Económicas Administrativas y Contables, Fundación Universitaria Católica del Norte, Medellín, Antioquía, Colombia, 5 Instituto de Investigación y Estudios de la Mujer, Universidad Ricardo Palma, Lima, Peru

* valenciajho@uss.edu.pe

## Abstract

The study on the adoption of mobile learning in university education reveals a growing interest in mobile technologies to improve the learning process; both the acceptance and rejection of these tools among students have been analyzed. However, there are gaps in the research that require a deeper exploration of the factors that influence the adoption and use of these technologies. Understanding these aspects is crucial to optimize mobile learning strategies and improve the educational experience in the university setting. The objective is to examine research trends regarding the topic. PRISMA-2020 is used in the Scopus and Web of Science databases. The results show the questionnaires as the main collection instruments; geographical contexts show that it has been researched predominantly in Asia; The studies have focused on university students; the most applied theories are TAM and UTAUT; and latent variables such as behavioral intention and attitude. The conclusions summarize the trends and patterns observed in the reviewed literature, as well as the research gaps identified, providing a solid foundation for future research and highlighting the importance of addressing this issue in the current context of digital education. The systematic review identifies key models and factors in the adoption of mobile learning in university settings, revealing both theoretical and practical implications. Furthermore, this text provides practical guidance for selecting effective data collection tools and making informed educational and policy decisions. However, it acknowledges limitations such as potential publication and language bias in the search process.

## 1. Introduction

The integration of mobile learning platforms in the university context is a significant topic of interest in contemporary educational research. With the growing prevalence of mobile devices and digital technologies, these platforms have been adopted to improve accessibility and flexibility of learning. Mobile learning is the use of mobile devices to facilitate the teaching and learning process. It has transformed educational dynamics by allowing students to access

**Data Availability Statement:** The data underlying the results presented in the study are available from https://doi.org/10.5281/zenodo.1065549

**Funding:** The author(s) received no specific funding for this work.

**Competing interests:** The authors have declared that no competing interests exist.

educational resources at any time and place [1]. This modality of education not only offers flexibility in terms of time and location but also provides opportunities for personalized learning, peer collaboration, and active student participation [2].

The acceptance and adoption of mobile learning among students and teachers is a crucial research topic that has generated a significant body of literature. Theoretical models have been proposed to understand the factors that influence students' intention to use mobile learning. These models highlight elements such as previous experience with technology, perceived usefulness, and ease of use [3]. Recent research has also explored how factors such as mobile self-efficacy and 21st-century skills influence the willingness of teachers to adopt mobile learning technologies in their pedagogical practices [4].

Empirical analysis has identified reasons and perceptions that influence the adoption of mobile learning applications among students [5]. Investigating the changing dynamics and emerging contexts in the use of mobile learning is crucial, particularly in light of external events such as the COVID-19 pandemic, to understand students' response to and experience with educational technologies [6].

The integration of mobile devices and digital technologies in university learning environments has made mobile learning an increasingly popular topic. This is due to its potential to improve the accessibility, flexibility, and effectiveness of learning. Mobile learning allows students to access educational resources conveniently and personalize their learning experience anytime and anywhere. Its adoption is important because it can transform traditional teaching methods and facilitate the creation of more dynamic and interactive learning experiences [7].

Recent research has investigated different aspects of mobile learning adoption in the university context. These studies have analyzed the factors that affect students' perception of mobile learning and its impact on enhancing learning. For instance, researchers have analyzed mobile learning adoption models that consider student perceptions as key determinants for improving the learning process [8]. Additionally, studies have identified socioeconomic and cultural factors that influence students' attitudes towards the use of mobile devices in learning, highlighting the importance of understanding contextual differences in the adoption of these technologies [9].

Understanding the factors that influence the adoption of mobile learning systems among university students is crucial for designing effective implementation and promotion strategies. Previous studies have examined the impact of theoretical models, such as the Technology Acceptance Model (TAM) and the SOR (Stimulus-Organism-Response) Model, on enhancing learning through mobile learning [10]. In 2023, the adoption of mobile learning systems in the Indonesian educational context was examined, emphasizing the significance of cultural and contextual factors in their implementation [11]. These studies underscore the importance of researching and understanding the adoption processes of mobile learning in the university context to optimize its potential as an educational tool.

The topic analyzed in this study has gaps that require attention and systematic analysis. Although various systematic reviews have been carried out in the field of mobile learning adoption, there is a need to delve into current trends and the factors that influence the acceptance and use of these technologies in specific university environments. For instance, while studies like Kumar and Chand [12] and Alsharida et al. [13] have explored the general adoption of mobile learning, further research is needed to examine how factors such as technostress and compatibility can impact the adoption of mobile learning among foreign language learners, as suggested by Wang et al. (insert year here). These gaps in the literature justify conducting a systematic review in 2022 that integrates and critically analyzes the available evidence. This will allow for the identification of emerging research areas and contribute to the theoretical and practical development of mobile learning adoption in specific university contexts. The

purpose of this study is to analyze research trends in the adoption of mobile learning in the university context from 2013 to 2024. The following questions will guide the research:

RQ1: What are the primary data collection instruments utilized in articles regarding the implementation of mobile learning in university settings?

RQ2: In what geographical contexts has the implementation of mobile learning in university settings been studied?

RQ3: What are the various population segments that have been the focus of research on the implementation of mobile learning in university settings?

RQ4: What psychobehavioral theories are used to understand the adoption of mobile learning in the university context?

RQ5: What are the primary latent variables or constructs used to comprehend the adoption of mobile learning in the university context?

This study compiles and synthesizes various theories, variables, and models used to understand the adoption of mobile learning in university educational environments. The aim is to identify predominant trends and approaches in research and offer a comprehensive vision of the factors that influence the acceptance and use of mobile learning in different university contexts worldwide. The study provides a solid foundation for building a unified mobile learning adoption model.

This study aims to identify the countries and populations that have been researched in this field. Recognizing geographical and demographic variations in the implementation and acceptance of this educational modality is important. The goal is to develop a conceptual framework based on the unified model that is applicable and relevant in various cultural and socioeconomic contexts. This integrative approach enables us to advance the theoretical understanding of mobile learning adoption in higher education and inform more effective and contextualized educational implications.

## 2. Methodology

Exploratory research was conducted using secondary sources. The methodology was based on the parameters and guidelines established by the PRISMA-2020 declaration, which provides a rigorous and transparent framework for conducting and presenting systematic reviews. Relevant studies were carefully selected, and key data were extracted to explore the factors that influence the adoption of mobile learning in specific university environments. This allowed for the identification of trends, research gaps, and areas of interest for future research in this emerging field of digital education.

### 2.1. Eligibility criteria

The eligibility criteria are divided into two sections. The first section includes inclusion criteria that mainly focus on titles and keywords as metadata. Specifically, it looks for the combination of terms such as 'mobile learning' and 'university' in various forms of citation, including variations such as 'm-learning' and 'mobile learning'. These criteria allow for an exhaustive and precise search for relevant studies that address the adoption of mobile learning in university environments, ensuring the inclusion of the most relevant literature for analysis.

The exclusion process involves three phases. The first phase excludes all records with erroneous indexing or those not directly related to the study's topic. The second phase of exclusion aims to eliminate all documents for which full text access is not available. This phase applies

only to Systematic Literature Reviews since the review in question focuses exclusively on the analysis of metadata. Finally, the third phase, the Exclusion phase, is responsible for discarding documents that do not present a clearly defined or explicit mobile learning adoption model. These exclusion criteria ensure the rigor and quality of the ongoing systematic literature review's study selection process.

## 2.2. Source of information

The Scopus and Web of Science databases were chosen as the primary sources of information. Scopus and Web of Science are considered the main bibliometric databases today due to their wide coverage and reputation in the academic and scientific fields. Research, such as that conducted by [14], has compared the quality and coverage of different bibliometric databases, concluding that Scopus and Web of Science are two of the most complete and reliable platforms available. Similarly, Tennant [15] conducted a study comparing the quality and coverage of different bibliometric databases, contributing to the understanding of the scope of platforms such as Scopus and Web of Science in the field of scientific knowledge collection. Although it is important to acknowledge that no database is entirely comprehensive, both Scopus and Web of Science provide a broad selection of academic and scientific journals, along with advanced search and analysis tools, making them ideal options for conducting a systematic literature review in a university setting.

## 2.3. Search strategy

To facilitate the search for relevant studies in the Scopus and Web of Science databases, two specialized search equations were designed. These equations were adapted to the defined inclusion criteria and the search characteristics of each platform. They were meticulously developed to ensure comprehensiveness and precision in identifying relevant articles on the adoption of mobile learning in the university context. The search equations were materialized on January 30, 2024, taking advantage of the advanced search functionalities of both databases to maximize the collection of relevant literature in the field of study.

For the Scopus database: (TITLE (("mobile learning") OR (mlearning) OR (m-learning)) AND TITLE (student OR scholar OR undergraduate OR learner) AND TITLE ((adoption) OR (use) OR (acceptance) OR tam OR tpb OR utaut)) OR (KEY (("mobile learning") OR (mlearning) OR (m-learning)) AND KEY (student OR scholar OR undergraduate OR learner) AND KEY ((adoption) OR (use) OR (acceptance) OR tam OR tpb OR utaut))

For the Web of Science database: (TI = (("mobile learning") OR (mlearning) OR (m-learning)) AND TI = (student OR scholar OR undergraduate OR learner) AND TI = ((adoption) OR (use) OR (acceptance) OR TAM OR TPB OR UTAUT)) OR (AK = (("mobile learning") OR (mlearning) OR (m-learning)) AND AK = (student OR scholar OR undergraduate OR learner) AND AK = ((adoption) OR (use) OR (acceptance) OR TAM OR TPB OR UTAUT))

## 2.4. Data management

The study utilized the Microsoft Excel® tool to extract, store, and process information from selected databases. This tool provided an organized structure to record relevant data from identified studies, allowing for efficient subsequent analysis. Each article obtained from the databases underwent an extensive and thorough full-text review to identify its relevance, contributions, and findings regarding the adoption of mobile learning in university settings. This systematic and detailed approach ensured completeness and quality in the collection and analysis of scientific literature relevant to the study's topic.

## 2.5. Selection process

Following the PRISMA 2020 statement guidelines, it is crucial to utilize internal automatic classifiers to facilitate the systematic literature review study selection process [16]. This practice helps to mitigate the risk of missing studies or incorrect classifications when applying inclusion and exclusion criteria more efficiently. Additionally, it is essential to validate these classifiers internally or externally to understand and control the risk of bias in study selection. In this study, we utilized an automation tool created in Microsoft Excel® as an internal classifier. All researchers involved in the study independently applied this tool during the study selection process, using predefined inclusion and exclusion criteria. This approach helped to minimize the risk of missing studies or incorrect classifications by converging the results and carefully reviewing the extracted metadata.

Furthermore, we used a specific Microsoft Excel® tool to homogenize all the articles extracted from both sources of information. This facilitated the process of excluding duplicates and applying the predefined inclusion and exclusion criteria. This ensures clear and unambiguous identification of the texts that will be analyzed in-depth for this systematic literature review. It guarantees consistency in the selection process mentioned earlier and contributes to the study's integrity and validity by minimizing the risk of missing relevant studies or making incorrect classifications.

## 2.6. Data collection process

As per the guidelines of [16], it is essential to specify the methods employed for collecting data from reports in a systematic literature review. In this study on the adoption of mobile learning in the university context, we used Microsoft Excel® as an automated tool for data collection from the selected databases, Scopus and Web of Science. The authors acted as reviewers for data validation, with each author conducting an independent evaluation to ensure an objective and thorough assessment of the information extracted from the studies. Subsequently, the authors collectively confirmed the data, comparing and contrasting the results obtained by each reviewer. The process was developed until achieving absolute convergence in the results, ensuring the reliability and integrity of the data collected in the literature review systematics.

## 2.7. Data elements

The objective was to gather data from all articles that met the research objective, which required adherence to the specialized search equation created for each database. This involved searching for results related to the implementation of mobile learning in the university context. The selected texts covered relevant measurements, time points, and analyses. However, if any information was missing or unclear, it was excluded as 'non-relevant texts' since they do not contribute to the understanding of knowledge on the topic. The purpose and scope of the research were considered to ensure consistency, allowing for the inclusion of significant and relevant results for the analysis of the adoption of mobile learning in the university context.

## 2.8. Assessment of the risk of bias of the study

The process of assessing the risk of bias in the included studies was a collaborative effort among all authors. The authors used the same automated Microsoft Excel® tool for data collection and evaluation of included studies. Each author independently assessed the studies using predefined criteria to identify potential sources of bias. The use of this automated tool standardized the evaluation process, ensuring the quality and integrity of the results. This

comprehensive and rigorous approach contributed to the validity and reliability of the systematic literature review on the adoption of mobile learning in the university context.

## 2.9. Measures of effect

It is relevant to specify that the effect measures traditionally used in primary research, such as the risk ratio or the difference in means, are not applicable in the analysis of secondary research sources. In this study, variables related to the data collection instruments, the geographical context of application of the study, the target population, the psychobehavioral theory used and latent variables within each evaluated model are analyzed. These aspects are addressed through the use of Microsoft Excel® to organize and analyze the data, as well as the use of VOSviewer® to determine thematic associations between the selected studies, this allows a deeper and more holistic understanding of the adoption of mobile learning in the university context, expanding the scope beyond conventional effect measures and providing a comprehensive view of the factors that influence this educational phenomenon.

## 2.10. Synthesis methods

It was established that all the studies included in the analysis had to be open access to ensure the availability of the full text and facilitate a thorough examination of each article. The data extracted from the selected studies were then stored in Microsoft Excel®. This tool provided a centralized platform to systematically tabulate and organize information, allowing for the comparison of study characteristics, preparation of data for presentation and synthesis, and efficient and coherent display of results. The use of Microsoft Excel® as a data management tool contributed to the rigorous organization and structured analysis of the information collected in this systematic review.

## 2.11. Assessment of reporting bias

When conducting a systematic literature review, it is important to be aware of potential biases towards certain synonyms found in thesauri, such as the IEEE. These biases may influence inclusion criteria, search strategy, and data collection, which could result in the exclusion of relevant studies that use alternative terms to describe the concept of mobile learning adoption. Additionally, excluding texts without a defined adoption model may lead to the omission of valuable information that could contribute to the understanding and construction of knowledge on the subject. Therefore, it is essential to take steps to mitigate the impact of these potential biases on the systematic literature review process.

## 2.12. Certainty evaluation

As part of this systematic investigation, we comprehensively and exhaustively evaluated the certainty of the body of evidence. We applied inclusion and exclusion criteria to each study to determine the suitability of the selected articles. Additionally, we conducted an individual evaluation of each article to identify any possible methodological biases or limitations of the study. These aspects were mentioned in both the description of the methodological designs and the discussion of the study's limitations. This contributed to a comprehensive evaluation of the certainty of the body of evidence, ensuring the transparency and reliability of the results obtained in the systematic review of literature on the adoption of mobile learning in university contexts (see Fig 1).

In this systematic literature review, the selection and exclusion of studies were carried out in several stages. First, we conducted an exhaustive search in selected information sources to

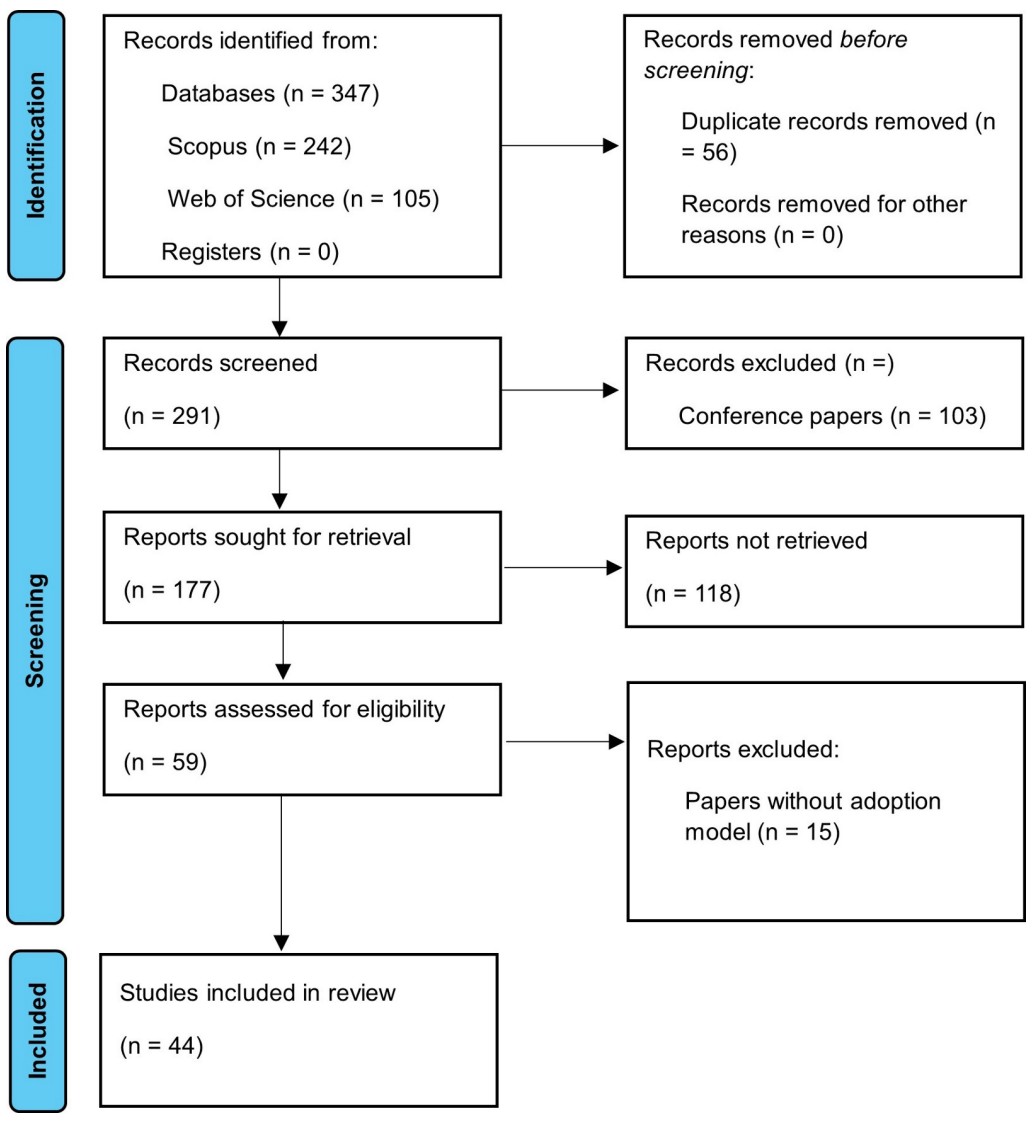

**Fig 1. PRISMA flowchart.** Own elaboration based on Scopus and Web of Science.

identify relevant studies. Then, we eliminated duplicate records to ensure data integrity. Finally, we proceeded with the selection and exclusion of studies based on pre-defined criteria. Three exclusion phases were carried out, applying predefined criteria to discard studies that did not meet the research objectives and scope. After this rigorous selection process, 44 articles were included as pertinent and relevant to address the issue of mobile learning adoption in the university context.

## 3. Results

The results section provides a comprehensive overview of the findings obtained from the systematic analysis of the relevant literature. This section presents the main emerging results of the review in an organized and structured manner. It addresses key aspects such as the instruments used for data collection, the geographical contexts where the phenomenon has been

studied, the population segments under investigation, the theoretical models used, and the latent variables or constructs identified.

This systematic literature review examines the adoption of mobile learning in university environments, following the parameters established by the PRISMA-2020 declaration. The summary of the articles included in the study is presented in Table 1, which includes only those that passed the inclusion phase and the three exclusion phases. This summary provides a clear and transparent understanding of the evidence base used in the analysis of the adoption of mobile learning in the university environment.

Table 2 details the identification and classification of the data collection instruments used in the included studies. These instruments are essential for understanding how information related to the adoption of mobile learning is collected in the university context. The analysis reveals that questionnaires and surveys are the main data collection instruments used by researchers to gather information on the adoption of mobile learning in university environments. This provides a clearer understanding of the phenomenon.

The study provides a comprehensive review of the geographical contexts in which various populations have been analyzed in relation to the adoption of mobile learning. Fig 2 presents these geographical contexts. The research highlights those Asian countries, including China, Turkey, Iran, India, Saudi Arabia, and Malaysia, among others, have been the most prominent in this field. Similarly, the topic has been extensively studied in Europe, with research conducted in countries such as Sweden, Spain, the United Kingdom, and Romania. This information offers a global perspective on the geographical distribution of research on mobile learning adoption in universities, highlighting worldwide areas of interest.

In addition to the geographical analysis, the study provides context about the population that the different authors have researched to understand the determining factors of mobile learning adoption. This context is presented in Fig 3. Research on the adoption of mobile learning in the university setting has primarily focused on university students in general, as well as students in various classifications. This suggests a broad and varied approach to understanding the topic. The information provides a clear vision of the interest groups in this field of research.

Fig 4 outlines the theoretical frameworks and psychobehavioral models utilized by researchers to forecast the factors that influence the adoption of mobile learning. The authors predominantly use the Technology Acceptance Theory (TAM) followed by others such as the Unified Model of Acceptance and Use of Technology (UTAUT), Proprietary Models, Extended UTAUT, and Theory of Planned Behavior (TPB). This highlights the diversity of theoretical approaches used in research on the adoption of mobile learning in the university setting, contributing to a more complete understanding of this phenomenon.

Fig 5 presents the main latent variables, factors, or constructs that different authors have adopted to understand the adoption of mobile learning among university populations in various geographical contexts. Researchers exploring the adoption of mobile learning in university settings have identified several key variables, including Behavioral Intention, Attitude, Expectation of Effort, Current Use, Compatibility, Confirmation, Academic Relevance, and Commitment. This information provides a deeper and more holistic understanding of the phenomenon.

## 4. Discussion

This section provides a detailed analysis of the research results, presenting their relevance and meaning. The findings are discussed, and the theoretical and practical implications derived from the results are presented. The study's limitations are also identified. The study identifies

**Table 1. Summary of the studies included in the systematic review.**

| N° | Title | Authors | Sample | Theory | Variables |
|---|---|---|---|---|---|
| 1 | A Model for Medical Students' Behavioral Intention to Use Mobile Learning | [1] | 376 | TPB | Perceived usefulness; Perceived ease of use; Attitude; Subjective norm; Perceived behavioral control; Perceived self-efficacy; Learning autonomy; Behavioral intention |
| 2 | A MODEL PREDICTING STUDENT ENGAGEMENT AND INTENTION WITH MOBILE LEARNING MANAGEMENT SYSTEMS | [2] | 253 | TAM; TTF; Own model | Commitment; Intention to continue use; Perceived usefulness; Perceived ease of use; Compatibility; Convenience; Enjoyment; Social influence; Self-efficacy; Personal innovation; Task-technology fit |
| 3 | A proposed framework to understand the intrinsic motivation factors on university students' behavioral intention to use a mobile application for learning | [17] | 193 | TAM | Perceived competence; Perceived challenge; Perceived choice; Perceived interest; Behavioral intention to use |
| 4 | Acceptance of Mobile Learning Technology by Teachers: Influencing Mobile Self-Efficacy and 21st-Century Skills-Based Training | [3] | 619 | UTAUT | Performance expectations; Effort expectations; Facilitating conditions; Social influence; Mobile self-efficacy; Student self-efficacy; Creative thinking skills; Training based on 21st century skills; Behavioral intention to use |
| 5 | Actual Use of Mobile Learning Technologies during Social Distancing Circumstances: Case Study of King Faisal University Students | [4] | 428 | UTAUT; D&M | Performance expectation; Expectation of effort; Social influence; Facilitating conditions; Intention to use; System quality; Quality of the information; Quality of service; User satisfaction; Actual use |
| 6 | An Empirical Investigation of Reasons Influencing Student Acceptance and Rejection of Mobile Learning Apps Usage | [5] | 415 | TAM; RTD; SCT | Perceived usefulness; Perceived ease of use; Perceived compatibility; Perceived convenience; Perceived self-efficacy; Perceived enjoyment; Behavioral intention to use |
| 7 | ANALYSIS OF A MOBILE LEARNING ADOPTION MODEL FOR LEARNING IMPROVEMENT BASED ON STUDENTS' PERCEPTION | [7] | 264 | Perceived interactivity model; TAM | Student-student interactivity; student-teacher interactivity; student-content interactivity; Perceived ubiquity; Perceived ease of use; Quality of learning content; Network quality; System quality; Perceived usefulness; Satisfaction; Perceived enjoyment; Continuity intention |
| 8 | Applications of mobile learning on college students' water leisure sports in the post-pandemic era | [8] | 370 | TAM | Perceived ease of use; Perceived usefulness; Intention to participate; Involvement in leisure; Benefits of leisure; Leisure barriers |
| 9 | Can COVID-19 pandemic influence experience response in mobile learning? | [6] | 627 | MTAM | Mobile Utility; Mobile Ease of Use; Quality of Learning Content; Interactivity; User interface; Connectivity; Experience Response |
| 10 | Determinants of Economical High School Students' Attitudes toward Mobile Devices Use | [9] | 407 | extended TAM | Network security; machine learning; Facilitating conditions; Self-efficacy; Perceived usefulness; Perceived usefulness; Intention to use mobile technologies |
| 11 | Determinants of the adoption of mobile learning systems among university students in Indonesia | [11] | 696 | Own model | Perceived mobility; Social influence; Self-efficacy; Personal innovation; Facilitating Condition; Learning Autonomy; Perceived enjoyment; Perceived usefulness; Perceived ease of use; Behavioral intention |
| 12 | E-mobile acceptance using unified theory of acceptance and use of technology (UTAUT): Research on universities in Jordan | [18] | 100 | UTAUT; Own model | Performance expectation; Expectation of effort; Social influence; Facilitating conditions; Culture; Educational factors |
| 13 | Enhancing students' English language learning via M-learning: Integrating technology acceptance model and S-O-R model | [10] | 1432 | TAM; SOR model | Perceived convenience; Perceived ease of use; Perceived usefulness; Attitude towards use; Intention to use; Curiosity; Self-efficacy |
| 14 | Examining the moderating role of technostress and compatibility in EFL Learners' mobile learning adoption: A perspective from the theory of planned behaviour | [19] | 409 | TPB; Compatibility; Technostress | Attitude; Perceived behavioral control; subjective norms; Intention; Technostress; Compatibility |

*(Continued)*

**Table 1.** (Continued)

| N° | Title | Authors | Sample | Theory | Variables |
|---|---|---|---|---|---|
| 15 | Exploring the factors affecting mobile learning for sustainability in higher education | [20] | 200 | TAM | Perceived usefulness; Perceived ease of use; Perceived enjoyment; task-technology fit; Perceived resources; Attitude towards use; Behavioral intention; Actual use of M-learning |
| 16 | Exploring the Technological Acceptance of a Mobile Learning Tool Used in the Teaching of an Indigenous Language | [21] | 68 | UTAUT | Perception of mobile device use; perceived usefulness; perceived ease of use; social influence; facilitating conditions; perceived entertainment |
| 17 | Factors Affecting Mobile Learning Acceptance in Higher Education: An Empirical Study | [22] | 218 | UTAUT; TAM | Relative advantage; Expectation of effort; Social influence; Facilitating conditions; Personal innovation; Self-management |
| 18 | Factors Affecting Mobile Learning Adoption in Healthcare Professional Students Based on Technology Acceptance Model | [23] | 310 | TAM | External factors; Perceived usefulness; Perceived ease of use; Attitude; Intention to use; Current usage |
| 19 | Factors Affecting Students' Acceptance of Mobile Learning Application in Higher Education during COVID-19 Using ANN-SEM Modelling Technique | [24] | 3000 | TAM | Content quality; System quality; Quality of service; Senior management support; Technological infrastructure; Perceived ease of use; Perceived usefulness; Behavioral intention; Current usage |
| 20 | Factors Influencing Students' Acceptance of M-Learning: An Investigation in Higher Education | [25] | 174 | UTAUT; Own model | Performance expectation; Expectation of effort; Social influence; Quality of service; Personal innovation; Behavioral intention |
| 21 | Factors influencing students' adoption and use of mobile learning management systems (m-LMSs): A quantitative study of Saudi Arabia | [26] | 258 | extended UTAUT | Performance expectation; Expectation of effort; Social influence; Facilitating conditions; Perceived mobile value; Academic relevance; Support from university management; Behavioral intention |
| 22 | Factors that affect University College Students' acceptance and use of Mobile Learning (ML) | [27] | 2077 | Own model | Influence of others; ML Effort; Benefits of ML; Reasons for use; Student capabilities; Usage expectations; Usage rate |
| 23 | Factors that influence students' acceptance of mobile learning for EFL in higher education | [28] | 332 | TAM | Perceived ease of use; Perceived usefulness; Attitude towards use; Behavioral intention |
| 24 | For sustainable application of mobile learning: An extended utaut model to examine the effect of technical factors on the usage of mobile devices as a learning tool | [29] | 612 | UTAUT | Performance expectation; Expectation of effort; Social influence; Price value; Device connectivity; Device Compatibility; Device security; reliability; Device processing power; Device memory capacity; Device performance; Network coverage; Network speed |
| 25 | Implementing effective learning with ubiquitous learning technology during coronavirus pandemic | [30] | 600 | UTAUT | Performance expectation; Expectation of effort; Social influence; Facilitating conditions; Behavioral intention |
| 26 | Integrating flipped foreign language learning through mobile devices: Technology acceptance and flipped learning experience | [31] | 84 | TAM; Own model | System caracteristics; Material characteristics; Perceived ease of use; Perceived usefulness; Attitude towards use; Behavioral intention; Motivation; Effectiveness; Commitment; Satisfaction |
| 27 | Investigating factors affecting on medical sciences students' intention to adopt mobile learning | [32] | 332 | TPB | Perceived ease of use; Perceived usefulness; Attitude; Subjective norm; Perceived behavioral control; Instructor disposition; Student disposition; Perceived self-efficacy; Learning autonomy; Intention |
| 28 | INVESTIGATING FACTORS THAT AFFECT THE CONTINUANCE USE INTENTION AMONG THE HIGHER EDUCATION INSTITUTIONS? LEARNERS TOWARDS A GAMIFIED M-LEARNING APPLICATION | [33] | 269 | NDE; UTAUT2 | Perceived usefulness; Perceived enjoyment; Facilitating conditions; Social influence; Perceived ease of use; Confirmation; Satisfaction; Intention for continued use |
| 29 | Investigating university students' intention to use mobile learning management systems in Sweden | [34] | 130 | TAM | Perceived mobility value; Academic relevance; University management support; Perceived usefulness; Perceived ease of use; Attitude towards use; Behavioral intention |

*(Continued)*

**Table 1.** (Continued)

| N° | Title | Authors | Sample | Theory | Variables |
|---|---|---|---|---|---|
| 30 | Learning from anywhere, anytime: Utilitarian motivations and facilitating conditions for mobile learning | [35] | 138 | TAM; UTAUT | Perceived ease of use; Perceived usefulness; Attitudes; Social influences; Facilitating conditions; Intentions; Stake |
| 31 | Mobile Learning Acceptance Post Pandemic: A Behavioural Shift among Engineering Undergraduates | [36] | 675 | C-TAM-TPB | Attitude (ATT); Behavioral Intention (BI); Mobile Learning Self-Efficacy (MSE); Perceived Ease of Use (PE); Perceived Utility (PU); Subjective Norm (SN) |
| 32 | Predicting mobile learning acceptance: An integrated model and empirical study based on the perceptions of higher education students | [37] | 246 | UTAUT extended with Expectancy-Confirmation Theory; Self-Determination Theory | Effort Expectation; Performance Expectation; Social influence; Facilitating Conditions; Confirmation; Perceived Autonomy; Perceived Competence; Behavioral Intention |
| 33 | Rural-based Science, Technology, Engineering and Mathematics teachers' and learners' acceptance of mobile learning | [38] | 288 | TAM | Behavioral Intention; Attitude towards Use; Perceived Utility; Perceived Ease of Use; Perceived Resources; Perceived Ease of Collaborating; Perceived Social Influence |
| 34 | Saudi Higher Education Student Acceptance of Mobile Learning | [39] | 683 | UTAUT | Performance Expectation; Effort Expectation; Social factors; Facilitating Conditions; Attitude towards Behavior; Behavioral Intention |
| 35 | Structural Determinants of Mobile Learning Acceptance among Undergraduates in Higher Educational Institutions | [40] | 415 | extended UTAUT | Performance Expectation; Effort Expectation; Social influence; Facilitating Conditions; Hedonic Motivation; Behavioral Intention |
| 36 | Structural Equation Modeling for Mobile Learning Acceptance by University Students: An Empirical Study | [41] | 384 | TAM; UTAUT | Perceived Utility; Perceived Ease of Use; Attitude; Social influence; Facilitating Conditions; Use of the Mobile Learning System |
| 37 | Student's Perception towards Mobile learning using Interned Enabled Mobile devices during COVID-19 | [42] | 1022 | TAM; UTAUT; DeLone; McLean | Information quality; Content quality; System quality; Service quality; Perceived ease of use; Perceived usefulness; Expectancy value theory; Satisfaction; Perceived mobility; Behavioral intention; Current use |
| 38 | Students' Perception of Cell Phones Effect on their Academic Performance: A Latvian and a Middle Eastern University Cases | [43] | 209 | TAM; IS continuance theory | Perceived usefulness; Perceived ease of use; attitude; Perceived enjoyment; Perceived mobility value; Behavioral intention; academic performance |
| 39 | Students' Perceptions of the Actual Use of Mobile Learning during COVID-19 Pandemic in Higher Education | [44] | 300 | TAM | Personal innovativeness; Task-technology fit; Perceived ease of use; Perceived usefulness; Students' satisfaction; Behavioral intention to use; Current use of mobile learning |
| 40 | The impact of COVID-19 on reading behaviors among high school students through the adoption of mobile learning | [45] | 394 | TAM; SCT; RTD | Perceived performance expectancy; Perceived effort expectancy; Perceived convenience; Self-efficacy; Perceived compatibility; Perceived enjoyment; COVID-19; Behavioral intention to use |
| 41 | The integrated social cognitive theory with the TAM model: The impact of M-learning in King Saud University art education | [46] | 412 | Integrated social cognitive theory; TAM | Social interaction; Social presence; Social spaces; Social identity; Perceived enjoyment; Perceived ease of use; Perceived usefulness; Behavioral intention to use; Student satisfaction; Current use |
| 42 | The study of ar-based learning for natural science inquiry activities in taiwan's elementary school from the perspective of sustainable development | [47] | 77 | TAM | Age; ICT experience; Digital literacy; Perceived usefulness; Perceived ease of use; Attitude towards use; Intention to use |
| 43 | Understanding Students' Intention to Use Mobile Learning at Universitas Negeri Semarang: An Alternative Learning from Home During Covid-19 Pandemic | [48] | 147 | TAM | Perceived ease of use; Perceived usefulness; Perceived interactivity; Satisfaction; Intention to use |
| 44 | Understanding the antecedents of intention for using mobile learning | [49] | 260 | Own model | Epistemological curiosity; Social influence; Security risk; Computer self-efficacy; Control locus; Perceived functional benefits; Student perceived value; Mobile learning adoption |

**Table 2. Data collection instruments.**

| Instrument | Frequency | Authors |
|---|---|---|
| Questionnaire | 39 | [1–11, 17, 18, 20, 22, 25–48] |
| Survey | 3 | [21, 23, 49] |
| Structured survey | 1 | [19] |
| Online survey | 1 | [24] |

the main research gaps and proposes a research agenda based on the results. Additionally, a theoretical model on the adoption of mobile learning in university environments is presented, utilizing the main theories and variables identified. This section is crucial for contextualizing and providing meaning to the review results, as well as guiding future research in the field of university mobile learning.

## 4.1. Analysis of data collection instruments

The results section indicates that the primary data collection instruments used were questionnaires and surveys. Several studies have contributed significantly to the understanding of questionnaires as data collection instruments. For example, Kumar et al. (2022) examined the behavioral change among university engineering students in the acceptance of mobile learning after the pandemic, providing an insightful view on how students perceive and adopt this educational modality. Similarly, Camilleri and Camilleri [35] explored the utilitarian motivations and facilitating conditions for mobile learning, delving into the factors that influence the adoption of this technology.

Baghcheghi et al. [23] analyzed the factors that affect the adoption of mobile learning in health professional students using the Technology Acceptance Model. The study provided

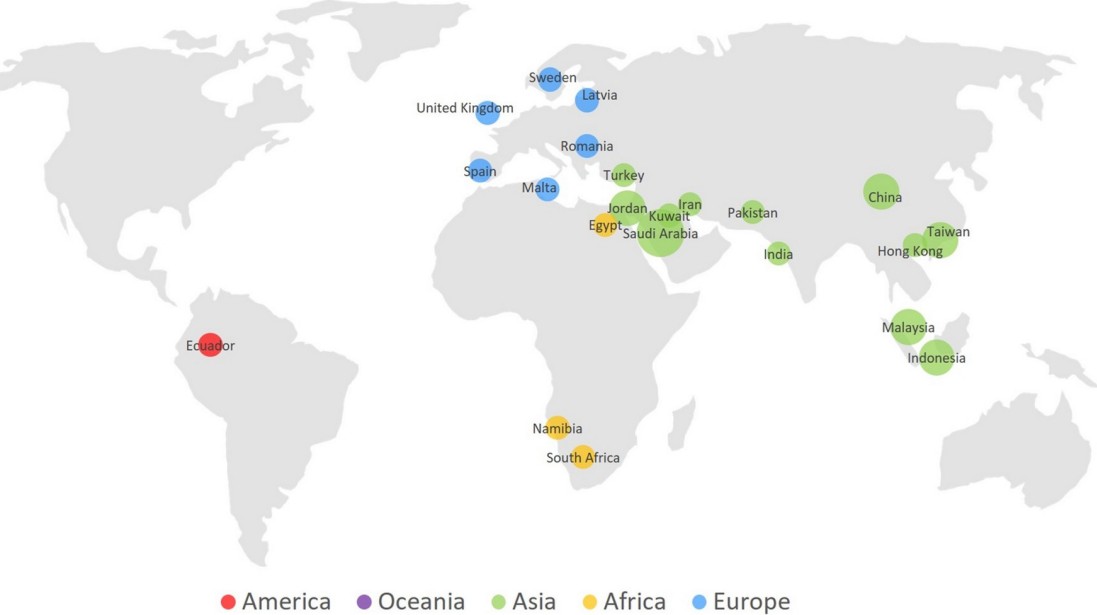

**Fig 2. Geographic context of the adoption of mobile learning in the university context.**

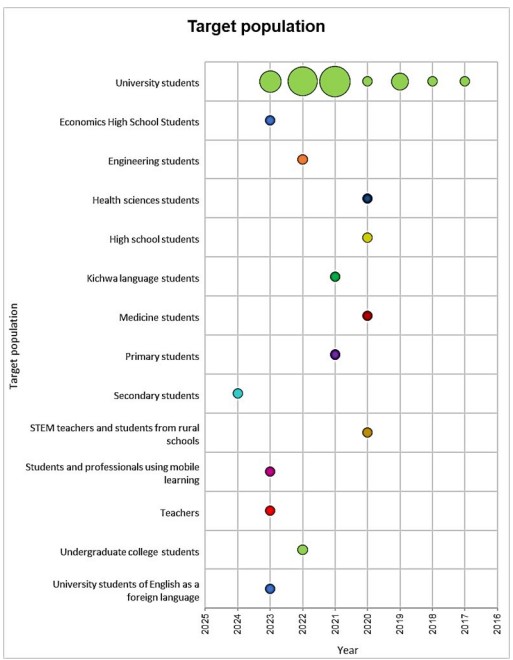

**Fig 3. Target population in the adoption of mobile learning in the university context.**

valuable information on students' perceptions and attitudes towards the use of mobile devices for learning. This research, along with others in the literature, has significantly contributed to the understanding of the factors that influence the adoption of mobile learning in the university setting. It has been a key reference in this field of research.

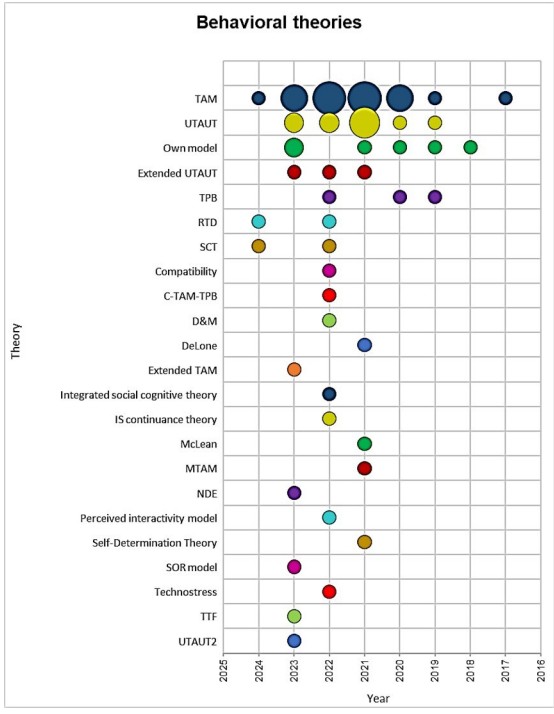

**Fig 4. Theories identified for the adoption of mobile learning in the university context.**

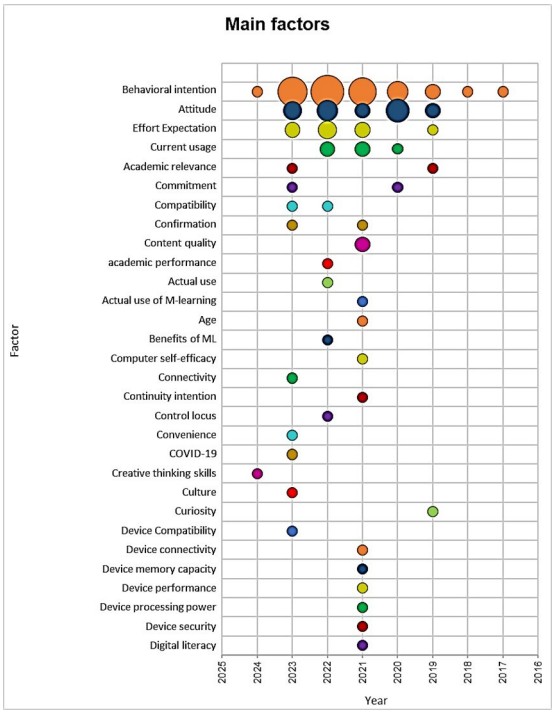

**Fig 5. Main variables of mobile learning adoption in the university context.**

## 4.2. Analysis of the geographical context of the adoption of mobile learning in the university context

The results section reveals that the theme has mainly occurred in Asia, including countries such as China, Turkey, Iran, India, Saudi Arabia, and Malaysia. It has also been observed in Europe, specifically in Sweden, Spain, the United Kingdom, and Romania. In China, Peng et al. [10] conducted a study on improving students' English language learning through mobile learning, integrating the Technology Acceptance Model and the SOR Model. Kucuk et al. [1] proposed a model for medical students' behavioral intention towards mobile learning in Turkey, examining their perceptions and attitudes in this educational field.

Azizi and Khatony [32] explored the factors that influence the intention of medical science students to adopt mobile learning in Iran, providing a detailed view of the variables that affect this decision. Gupta et al. [42] investigated Indian students' perception of mobile learning as a tool for education during the COVID-19 pandemic. Alturki and Aldraiweesh [44] analyzed the use of mobile learning in higher education during the pandemic in Saudi Arabia, providing valuable insights into students' experiences.

Saroia and Gao [34] investigated university students' intention to use mobile learning management systems in Sweden, exploring their attitudes and perceptions towards this emerging technology. Additionally, Andujar et al. [31] examined the integration of flipped learning through mobile devices in Spain, exploring technological acceptance and the flipped learning experience. Abu-Al-Aish and Love [25] investigated the factors that influence the acceptance of mobile learning among students in the United Kingdom. Their study provides a detailed view of the elements that affect the adoption of this educational modality.

### 4.3. Analysis of the target population in the adoption of mobile learning in the university context

As previously mentioned, the topic has gained prominence in several countries including Jordan, Malaysia, Saudi Arabia, China, Spain, India, and Indonesia. The results section indicates that research on this topic has primarily focused on university students and students in general in these countries. Ismiyati et al. [48] conducted a study to investigate Semarang State University students' intention to use mobile learning as an alternative to in-person learning during the COVID-19 pandemic. Alturki and Aldraiweesh [44] examined students' perceptions of the actual use of mobile learning in higher education during the pandemic, providing a detailed view of their experiences.

Lo et al. [47] conducted a study on augmented reality-based learning for natural science inquiry activities in primary schools in Taiwan, from the perspective of sustainable development. The study focused on students in general. Research sheds light on the effectiveness and sustainability of AR-based learning in the Taiwanese school context. These investigations represent significant contributions to understanding the adoption and benefits of mobile learning in different educational contexts.

### 4.4. Analysis of psychometric theories in the adoption of mobile learning in the university context

The results section reveals that the main theories used to understand the factors that determine the adoption of mobile learning in the university context are TAM, UTAUT, Own Models, Extended UTAUT, and TPB. Within the scope of Technology Acceptance Theory (TAM), Almaiah et al. [24] examined the factors that affect the acceptance of a mobile learning application in higher education during the COVID-19 pandemic, using the Ann-Sem modeling technique. Alghazi et al. [29] developed an extended model to examine the effect of technical factors on the sustainable use of mobile devices as a learning tool, based on the Unified Technology Acceptance Model (UTAUT).

Pramana [11] investigated the determinants of mobile learning system adoption among university students in Indonesia. Similarly, Alfalah [26] explored the factors influencing the adoption and use of mobile learning management systems among students in Saudi Arabia. Azizi and Khatony [32] investigated the factors that affect students' intention to adopt mobile learning in medical sciences, using the Extended UTAUT model and the Theory of Planned Behavior (TPB). The study provides a deep understanding of the underlying theories that influence the adoption of mobile learning in the university context.

### 4.5. Analysis of the main variables of adoption of mobile learning in the university context

The results indicate that the main latent variables used to determine adoption of mobile learning in the university context are Behavioral Intention, Attitude, Expectation of Effort, and Current Use. Regarding the Behavioral Intention variable, Andujar et al. [31] explored the integration of foreign language learning through mobile devices, focusing on technological acceptance and the flipped learning experience. As for the Attitude variable, Azizi and Khatony [32] investigated the factors that affect medical science students' intention to adopt mobile learning, analyzing the influence of their attitude towards this educational modality.

Dahri et al. [3] investigated teachers' acceptance of mobile learning technology, focusing on the influence of mobile self-efficacy and training based on 21st century skills, while Almaiah et al. [24] examined the factors affecting the adoption of a mobile learning application in

higher education during the COVID-19 pandemic using the Ann-Sem modeling technique. These studies provide a deeper understanding of the latent variables that influence the adoption of mobile learning in the university environment.

In addition to the previously mentioned variables, other factors have emerged as significant in analyzing the adoption of mobile learning in university contexts. One such factor is Academic Relevance, which pertains to students' perception of the usefulness and relevance of mobile learning for their academic training. The influence of the perception of usefulness and academic relevance on students' intention to use learning management systems has been explored in studies such as Alfalah [26] and Saroia and Gao [34]. This study examines the use of mobile learning management systems in different university environments.

Another important factor to consider is engagement, which refers to the level of dedication and emotional connection that students have with mobile learning. This variable has been analyzed by Imlawi et al. [2] and Andujar et al. [31] to understand how student engagement influences their intention to use mobile learning management systems in university environments. It is important for the active involvement of students in the educational process.

Compatibility is a relevant variable that has been extensively studied in the context of mobile learning adoption. It refers to students' perception of the agreement between mobile learning and their needs, skills, and technological environment [29]. This variable has been explored in studies such as those by Wang, Zhao, and Cheng [19] and Alghazi et al. The language used in the text is clear, concise, and objective, with a formal register and precise word choice. The text follows a logical structure with causal connections between statements. The grammar, spelling, and punctuation are correct. No changes were made to the content of the original text. Masa'deh et al. [45] and Chen [8] have investigated the impact of perceived compatibility on students' adoption of mobile learning. They analyzed how technical factors, such as technological stress and available resources, affect students' perceptions of mobile learning. It is important to consider the compatibility of mobile learning with your individual learning needs and expectations.

Finally, confirmation is another variable that has gained importance in the literature on the adoption of mobile learning in higher education. It is defined as the continuous evaluation that students make of the usefulness and effectiveness of mobile learning after its initial implementation. Mobile learning applications have been extensively studied to explore the factors that influence students' confirmation of continuing to use them. These studies, such as those by Roslan et al. and Alowayr and Al-Azawei, highlight the impact of these factors on user satisfaction and intention to continue using these technologies. The language used is clear, objective, and value-neutral, with a formal register and precise word choice. The text follows conventional structure and adheres to formatting features and style guides. The grammar, spelling, and punctuation are correct. No changes in content have been made.

From another perspective, it is crucial to take into account the perspectives provided by Al-Adwan, Al-Adwan and Berger [50] and Al-Adwan, Al-Madadha and Zvirzdinaite [51], highlighting the importance of analyzing other factors that influence the adoption of mobile learning in higher education, highlighting the relevance of delving into factors such as student disposition and the enigmatic nature of adoption to unravel the complexities surrounding the adoption of mobile learning.

### 4.6. Main research gaps

Table 3 presents the main research gaps identified in the field of mobile learning in the university context that need to be addressed in future research. These gaps highlight areas where the

**Table 3. Main research gaps identified.**

| Category | Gap | Justification | Questions for future research |
|---|---|---|---|
| Geographic gaps | Lack of studies in Latin American countries | Most studies on mobile learning adoption in university contexts focus on Asia and Europe, leaving a gap in the understanding of adoption in Latin American countries. | What are the specific factors that influence the adoption of mobile learning in universities in Latin America? How do these factors differ from those identified in other geographic contexts? |
| Theoretical Gaps | Little application of emerging theories | The literature on mobile learning adoption in university contexts often relies on traditional theories such as TAM and UTAUT, leaving aside emerging theories that could provide a more holistic understanding of the phenomenon. | How do emerging theoretical models compare to traditional models in terms of explaining the adoption of mobile learning in university contexts? What new aspects can emerging theories contribute to the understanding of the phenomenon? |
| Variable gaps | Little consideration of accessibility | Many studies on mobile learning adoption in university environments do not adequately consider the accessibility of mobile technologies, which could limit their application in various contexts. | How does the accessibility of mobile technologies influence the adoption of mobile learning in different groups of university students? What strategies can be implemented to improve the accessibility of mobile learning platforms? |

existing literature may be insufficient or where greater depth is needed to fully understand the adoption of mobile learning and its implications in the university context.

The identified gaps in research on the adoption of mobile learning in university contexts provide an opportunity for further exploration to enhance our understanding of this dynamic. One of the gaps identified is the lack of studies in Latin American countries, which highlights the need for specific investigation into the factors that influence the adoption of mobile learning in universities in this region. How do these factors differ from those identified in other geographic contexts? Examining these differences could provide insight into the cultural and socio-economic factors that impact the adoption of these technologies in various regions of the world.

Additionally, there is a lack of application of emerging theories in the study of mobile learning adoption. While traditional theories like TAM and UTAUT have been extensively utilized, incorporating emerging theories could provide a more comprehensive and current understanding of the phenomenon. How do these new theoretical models compare to traditional ones in explaining the adoption of mobile learning in university contexts? Investigating this question could reveal new facets of the adoption process and provide innovative insights for the design of implementation and promotion strategies.

Another significant gap identified is the lack of consideration of accessibility in studies on mobile learning adoption. Accessibility is a crucial factor in ensuring the effectiveness and equity of technologies for all students. To design more inclusive and effective strategies, it is important to explore how the accessibility of mobile technologies affects the adoption of mobile learning among diverse student groups. What strategies can be implemented to improve the accessibility of mobile learning platforms? Investigating this question could lead to identifying practices and policies that encourage broader and more equitable adoption of mobile learning in university settings.

The gaps in research on the adoption of mobile learning in university settings not only highlight areas where current knowledge is limited but also point to the importance of addressing these gaps to promote more complete and balanced development in this field. By exploring and closing these gaps, we can advance our theoretical understanding of mobile learning adoption. This will enable us to more accurately inform educational policies and practices that encourage the effective and equitable integration of these technologies into teaching and learning at universities.

## 4.7. Theoretical implications

The evaluation of the data collection instruments used in the studies allows us to identify trends, methodological approaches, and possible biases in the measurement of key variables.

Additionally, analyzing the geographical context of each study reveals regional patterns in the implementation and acceptance of mobile learning. This may suggest the influence of cultural, economic, and technological factors in the adoption of this educational modality.

Considering the target population of the studies provides valuable information on the demographic, academic, and socioeconomic characteristics of the university students involved in the adoption of mobile learning. Additionally, exploring the theoretical models used to understand the adoption phenomenon offers insights into the predominant theoretical perspectives and their applications in different educational contexts. Identifying the key factors used to approach the understanding of mobile learning in the university environment allows for a critical evaluation of the determinants that influence adoption and the effective use of this technology.

The systematic literature review, conducted using the PRISMA-2020 methodology, reveals the research gaps in the adoption of mobile learning in university contexts. Identifying gaps in research is crucial for future studies on mobile learning adoption in universities. These gaps may be due to a lack of research in certain geographic areas, scarcity of studies using emerging theories, absence of consideration of relevant factors, or the need to delve into specific aspects of the phenomenon. By identifying these gaps, efforts can be focused on areas where greater theoretical and empirical development is required to comprehensively understand mobile learning adoption in the university environment.

## 4.8. Practical implications

The current study has significant practical implications for both academics and decision-makers in the field of education. The evaluation of data collection instruments enables identification of the most effective ones for capturing relevant information on the adoption of mobile learning. This can guide academics in designing future research and developing evaluation tools for more accurate monitoring and measuring of progress in implementing this technology.

The analysis of the geographical context of each study provides a global view of the trends and specific challenges associated with the adoption of mobile learning in different regions of the world. This information is invaluable for decision-makers in the educational field as it allows them to identify geographic areas where greater support and resources are needed to promote the successful adoption of this educational modality, as well as to adapt implementation strategies to the local needs and realities of each context.

Consideration of the target population is crucial for understanding the specific characteristics and needs of university students in relation to mobile learning. This insight allows decision-makers to design educational programs and policies that best suit the preferences and abilities of the students, promoting greater participation and commitment to this learning modality.

The analysis of theoretical models and factors used to understand the adoption of mobile learning in the university context provides academics and decision-makers with a solid conceptual framework to design intervention strategies and training programs that encourage successful and sustainable adoption of this technology. Identifying research gaps is also essential as it highlights areas where further research and development are required to address specific challenges and maximize the impact of mobile learning in university education.

It provides educators and educational policy makers with a deep understanding of the trends, challenges, and best practices related to the integration of mobile learning in university environments. This allows them to make informed decisions about the implementation of

educational technologies and design teaching strategies that make the most of the potential of mobile learning to improve the learning experience of students.

Furthermore, a systematic review can impact the allocation of resources and strategic planning in educational institutions. It identifies priority areas for investment in technological infrastructure, teacher training, and development of digital content. Additionally, it can guide the formulation of policies and support programs that promote equity of access and digital inclusion, particularly for students who may face socioeconomic or geographic barriers to accessing digital educational resources.

Understanding the factors that influence the adoption and effective use of mobile learning in university environments can inform training and professional development strategies in companies and organizations. This can help design mobile learning programs that align with the needs and expectations of today's workforce. The practical implications can be extended to the industry and business sector.

Additionally, the review's findings and recommendations can inform government decision-makers in formulating education and technology-related public policies. This can promote effective integration of mobile learning into national or regional educational systems, fostering innovation and continuous improvement in higher education.

## 4.9. Limitations

One limitation of this systematic literature review is the potential for publication bias, as only studies available in the Scopus and Web of Science databases were included. It is possible that relevant studies not indexed in these databases or available in other languages were not considered, leading to a limited selection of literature. Additionally, restricting articles to English may have excluded significant research conducted in other languages, potentially biasing the results towards a specific linguistic perspective.

Another limitation of this study is related to the search process. Although we used broad search criteria and explored multiple combinations of terms related to mobile learning adoption in university environments, it is possible that some relevant studies were not identified due to the complexity and diversity of the terminology used in this field. Additionally, the exclusion of studies not available in full text may have limited the inclusion of relevant research that was only available in abstract format or with restricted access. These limitations may have affected the exhaustiveness and representativeness of the systematic review, which could impact the generalization of the results and conclusions obtained.

Finally, a limitation of this study is that important databases, such as ERIC, were excluded. ERIC is recognized as one of the main sources of information in the field of education. The omission of this database may have resulted in a lack of access to relevant studies that could have further enriched the analysis and understanding of the topic. Therefore, future studies should aim to include a wider range of databases to ensure a comprehensive review of the literature and a more complete representation of available research.

## 4.10. Agenda for future research

Several recommendations for future research can be derived from the obtained results, which could enhance the understanding of this emerging field. Firstly, longitudinal studies are suggested to track the evolution of mobile learning adoption over time and assess its long-term impact on academic achievement and the student experience. These investigations could provide a more complete understanding of how attitudes and behaviors towards mobile learning change over time and identify predictors of sustained adoption.

Furthermore, it is recommended to expand the geographical scope of research to include less explored contexts, such as Africa, Latin America, and Eastern Europe. Comparative studies between different geographical regions could help identify common patterns and significant differences in the factors that influence mobile learning adoption.

To improve the impact of mobile learning on diverse university student populations, it is recommended to conduct research that analyzes the needs and preferences of different demographic groups, including students from various disciplines, educational levels, and socioeconomic backgrounds. This will enable the design of more personalized interventions that are better adapted to the needs of each group.

Regarding theoretical models, researchers are encouraged to validate new conceptual frameworks that accurately capture the underlying processes that influence mobile learning adoption. It is also recommended to integrate multiple theoretical models and approaches to obtain a more holistic and multidimensional understanding of this complex phenomenon.

Finally, it is recommended to investigate emerging and under-researched variables that may impact the adoption of mobile learning, such as digital accessibility, technological inclusion, data privacy, and cybersecurity. These aspects are crucial to ensure fair and sustainable adoption of mobile learning in university environments and can lead to new areas of research that address emerging needs and concerns in this constantly evolving field.

### 4.11. Main adoption model of mobile learning in the university context

Fig 6 shows the main theoretical models and variables used to understand or predict the adoption of mobile learning in the university context. The TAM and the UTAUT are consolidated conceptual frameworks that have been widely used to understand the attitudes, perceptions, and behaviors of university students towards the use of mobile technologies in their educational processes. Theoretical models, along with associated variables such as perceived ease of use, perceived usefulness, attitude toward use, social influence, and intention to use, have been crucial in contextualizing and analyzing the adoption of mobile learning in various university environments.

This systematic literature review examines the adoption of Mobile Learning in the university context and proposes a comprehensive model that combines the UTAUT Model and the TAM. The model incorporates external variables for a holistic understanding of Mobile Learning adoption in university environments.

The proposed model integrates additional variables, such as Academic Relevance, Confirmation, and Compatibility factors, into the conceptual pillars of UTAUT and TAM. These variables are of vital importance in the academic field as they evaluate the relevance of Mobile Learning in the university educational context and determine the acceptance and effective use of this technology. This theoretical contribution provides a comprehensive and contextualized analytical framework for understanding the determinants that influence the adoption of Mobile Learning in the university environment. It can aid in the formulation of informed and relevant strategies in the higher education field.

## 5. Conclusions

The research has produced significant conclusions that address the research questions. The analysis indicates that questionnaires are the primary data collection instruments used in the studies, indicating a preference for quantitative methods to gather information on the adoption of mobile learning.

In terms of geographical contexts, research in the field of mobile learning has been primarily focused on Asia and Europe. Countries such as Saudi Arabia, China, the United Kingdom,

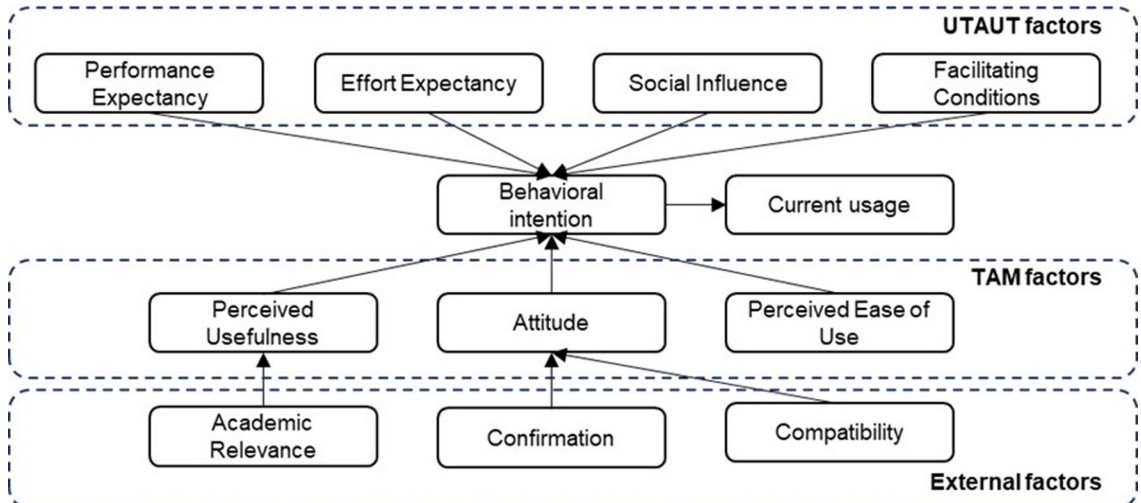

**Fig 6. Proposed theoretical model on adoption of mobile learning in the university context.**

and Spain have been leading this research. This finding emphasizes the global nature of mobile learning adoption and the importance of considering diverse contexts in future research.

Research on mobile learning adoption focuses mainly on university students, highlighting the importance of understanding the needs and perceptions of this demographic group when integrating mobile technologies in education. The identified theoretical models were TAM, UTAUT, and own models. This highlights the importance of understanding users' attitudes and perceptions towards mobile learning from a consolidated theoretical framework.

The main variables used to understand the adoption of mobile learning in university contexts are behavioral intention, attitude, effort expectation, current use, and compatibility. The article highlights the need to develop integrative theoretical models that address the factors influencing the adoption of mobile learning. It is also recommended to explore new variables and geographical contexts to enrich the understanding of the phenomenon.

Regarding future research, it is recommended to investigate the impact of mobile learning in various fields of study and educational contexts. Additionally, it is important to examine the influence of contextual and cultural factors on the adoption of these technologies.

A theoretical model is presented that integrates the main theoretical models and variables identified in the review, providing a conceptual structure for future research in the field of mobile learning adoption in the university context.

## Supporting information

**S1 Checklist. PRISMA 2020 checklist.**
(DOCX)

## Author Contributions

**Conceptualization:** Alejandro Valencia-Arias, Rosa María Vélez Holguín.

**Data curation:** Alejandro Valencia-Arias, Sebastian Cardona-Acevedo.

**Formal analysis:** Sergio Gómez-Molina.

**Investigation:** Sebastian Cardona-Acevedo, Jackeline Valencia.

**Methodology:** Sebastian Cardona-Acevedo, Rosa María Vélez Holguín.

**Resources:** Sergio Gómez-Molina.

**Software:** Sergio Gómez-Molina, Jackeline Valencia.

**Validation:** Rosa María Vélez Holguín.

**Visualization:** Alejandro Valencia-Arias, Jackeline Valencia.

**Writing – original draft:** Sebastian Cardona-Acevedo, Rosa María Vélez Holguín.

**Writing – review & editing:** Alejandro Valencia-Arias, Sebastian Cardona-Acevedo, Sergio Gómez-Molina, Rosa María Vélez Holguín, Jackeline Valencia.

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
