## [Decision Letter · Decision Letter 0]

20 Mar 2024

PONE-D-24-05965Adoption of mobile learning in the university context: systematic literature reviewPLOS ONE

Dear Dr. Valencia-Arias,

Thank you for submitting your manuscript to PLOS ONE. After careful consideration, we feel that it has merit but does not fully meet PLOS ONE’s publication criteria as it currently stands. Therefore, we invite you to submit a revised version of the manuscript that addresses the points raised during the review process.

**ACADEMIC EDITOR: **

For acceptance of the manuscript, the authors should address the following specific issues raised by the three reviewers:

Authors should prioritize addressing all relevant points raised by all three reviewers.Improve clarity of the paper: The authors should ensure that the abstract clearly highlights the implications of their findings and the introduction should emphasize the paper's innovation and contribution.Deepening analysis: The authors should conduct a more comprehensive analysis of factors influencing mobile learning adoption, as suggested by Reviewer 2.Clarify the methodology: The authors should provide additional clarity on the methodology, particularly regarding database selection and the selection process, as suggested by Reviewer 3.Improve presentation: The authors should enhance the resolution of figures to improve their quality, as recommended by Reviewer 3.==============================

We look forward to receiving your revised manuscript.

Kind regards,

Eric Amankwa, Ph.D.

Academic Editor

PLOS ONE

Journal Requirements:

2. We note that Figure 2  in your submission contain map image which may be copyrighted. All PLOS content is published under the Creative Commons Attribution License (CC BY 4.0), which means that the manuscript, images, and Supporting Information files will be freely available online, and any third party is permitted to access, download, copy, distribute, and use these materials in any way, even commercially, with proper attribution. For these reasons, we cannot publish previously copyrighted maps or satellite images created using proprietary data, such as Google software (Google Maps, Street View, and Earth). For more information, see our copyright guidelines: http://journals.plos.org/plosone/s/licenses-and-copyright.

Reviewers' comments:

Reviewer's Responses to Questions

**Comments to the Author**

1. Is the manuscript technically sound, and do the data support the conclusions?

Reviewer #1: Yes

Reviewer #2: Yes

Reviewer #3: Yes

2. Has the statistical analysis been performed appropriately and rigorously? 

Reviewer #1: I Don't Know

Reviewer #2: N/A

Reviewer #3: N/A

3. Have the authors made all data underlying the findings in their manuscript fully available?

Reviewer #1: Yes

Reviewer #2: Yes

Reviewer #3: Yes

4. Is the manuscript presented in an intelligible fashion and written in standard English?

Reviewer #1: Yes

Reviewer #2: Yes

Reviewer #3: Yes

5. Review Comments to the Author

Reviewer #1: In the abstract section, it would be valuable to include a sentence discussing the implications of identified reseaech trends and patterns, as well as a brief acknowledgement of the study limitations to provide transparency.

I think the authors have done a good job. However, I think they should clearly show the reader how their work differs from others in terms of stating the paper's innovation and contribution.

Reviewer #2: Thank you very much. This is an interesting paper. However, there are few concerns that need to be addressed. 1) it is important to highlight the main contributions of this paper at the introduction section, 2) The study provides valuable insights into the adoption of mobile learning in university education, shedding light on both acceptance and rejection factors among students. However, there is a need for a deeper exploration of the various factors influencing the adoption and usage of these technologies. A more comprehensive analysis of these aspects would contribute significantly to optimizing mobile learning strategies and enhancing the overall educational experience. 3) The conclusions drawn from the reviewed literature effectively summarize the observed trends and patterns while also identifying crucial research gaps. These findings serve as a robust foundation for future research endeavors, emphasizing the significance of addressing these gaps in the current landscape of digital education. Further exploration of these research gaps is essential to advancing our understanding of mobile learning adoption dynamics. 4) Additional studies should be included. This includes but not limited:

- Solving the mystery of mobile learning adoption in higher education. doi: https://doi.org/10.1504/IJMC.2018.088271

- Modeling Students’ Readiness to Adopt Mobile Learning in Higher Education: An Empirical Study. doi: https://doi.org/10.19173/irrodl.v19i1.3256

Reviewer #3: Good morning,

Thank you very much for the opportunity to read the article titled "Adoption of mobile learning in the university context: systematic literature review." The article addresses a current global issue. In fact, many European countries are considering limiting the use of mobile devices in teaching-learning processes.

Overall, the article is written in a clear and coherent manner, and it presents information structured in a way that facilitates understanding. Additionally, the theoretical references are up-to-date.

However, the methodology proposed has some elements that are not entirely clear. For example, I believe it would be interesting to explain why only two databases have been included. Why was ERIC not included, considering it is one of the main databases in education? Also, although the PRISMA methodology is mentioned for data selection, I would like to know more about the selection process.

Regarding the results, they are relevant. However, I missed a more quantitative section summarizing the overall view of the studies in the sub-sections. This way, it could complement the qualitative information presented with a quantitative overview of the studies.

In line with this, the figures are relevant in the selected sections. However, their quality is somewhat low. It would be interesting to avoid pixelation upon enlargement by improving the resolution.

Finally, the conclusions and limitations presented are relevant and coherent.

Good job.

6. PLOS authors have the option to publish the peer review history of their article (what does this mean?). If published, this will include your full peer review and any attached files.

Reviewer #1: No

Reviewer #2: No

Reviewer #3: No

---

## [Author Response · Author response to Decision Letter 0]

8 Apr 2024

Please find attached the file named "Response Letter" in response to each of the requests made by the reviewers and the editor. Additionally, Figure 1 and Table 3 are mentioned in the main text, according with the suggestions.

---

## [Decision Letter · Decision Letter 1]

7 May 2024

Adoption of mobile learning in the university context: systematic literature review

PONE-D-24-05965R1

Dear Dr. Valencia-Arias,

We’re pleased to inform you that your manuscript has been judged scientifically suitable for publication and will be formally accepted for publication once it meets all outstanding technical requirements.

Kind regards,

Eric Amankwa, Ph.D.

Academic Editor

PLOS ONE

Additional Editor Comments (optional):

Reviewers' comments:

Reviewer's Responses to Questions

**Comments to the Author**

1. If the authors have adequately addressed your comments raised in a previous round of review and you feel that this manuscript is now acceptable for publication, you may indicate that here to bypass the “Comments to the Author” section, enter your conflict of interest statement in the “Confidential to Editor” section, and submit your "Accept" recommendation.

Reviewer #1: All comments have been addressed

Reviewer #2: All comments have been addressed

Reviewer #3: (No Response)

2. Is the manuscript technically sound, and do the data support the conclusions?

Reviewer #1: Yes

Reviewer #2: Yes

Reviewer #3: Yes

3. Has the statistical analysis been performed appropriately and rigorously? 

Reviewer #1: I Don't Know

Reviewer #2: N/A

Reviewer #3: Yes

4. Have the authors made all data underlying the findings in their manuscript fully available?

Reviewer #1: No

Reviewer #2: Yes

Reviewer #3: Yes

5. Is the manuscript presented in an intelligible fashion and written in standard English?

Reviewer #1: Yes

Reviewer #2: Yes

Reviewer #3: Yes

6. Review Comments to the Author

Reviewer #1: (No Response)

Reviewer #2: Thank you for submitting the revised version. The quality has increased significantly after addressing the reviewers' comments.

Reviewer #3: Dear authors,

Firstly, I would like to express my gratitude for providing me with the opportunity to review the article titled "Adoption of mobile learning in the university context: systematic literature review." It was indeed a privilege to engage with such a timely and important piece of research.

Secondly, taking into account the modification made by the authors, the article is considered ready for publication.

Congratulations on the work.

7. PLOS authors have the option to publish the peer review history of their article (what does this mean?). If published, this will include your full peer review and any attached files.

Reviewer #1: No

Reviewer #2: No

Reviewer #3: No

---

## [Editor Report · Acceptance letter]

13 May 2024

PONE-D-24-05965R1 

PLOS ONE

Dear Dr. Valencia-Arias, 

I'm pleased to inform you that your manuscript has been deemed suitable for publication in PLOS ONE. Congratulations! Your manuscript is now being handed over to our production team.

Kind regards, 

on behalf of

Dr. Eric Amankwa 

Academic Editor

PLOS ONE